# The Effect of Resection Margin on Local Recurrence and Survival in High Grade Soft Tissue Sarcoma of the Extremities: How Far Is Far Enough?

**DOI:** 10.3390/cancers12092560

**Published:** 2020-09-08

**Authors:** Annika Bilgeri, Alexander Klein, Lars H. Lindner, Silke Nachbichler, Thomas Knösel, Christof Birkenmaier, Volkmar Jansson, Andrea Baur-Melnyk, Hans Roland Dürr

**Affiliations:** 1Musculoskeletal Oncology, Department of Orthopaedic Surgery, Physical Medicine and Rehabilitation, University Hospital, LMU 80539 Munich, Germany; anni.bilgeri@live.de (A.B.); alexander.klein@med.uni-muenchen.de (A.K.); Christof.Birkenmaier@med.uni-muenchen.de (C.B.); volkmar.jansson@med.uni-muenchen.de (V.J.); 2Department of Medicine III, University Hospital, LMU 80539 Munich, Germany; Lars.Lindner@med.uni-muenchen.de; 3Department of Radiation Oncology, University Hospital, LMU 80539 Munich, Germany; silke.nachbichler@med.uni-muenchen.de; 4Institute of Pathology, University Hospital, LMU 80539 Munich, Germany; Thomas.Knoesel@med.uni-muenchen.de; 5Department of Radiology, University Hospital LMU 80539 Munich, Germany; Andrea.Baur@med.uni-muenchen.de

**Keywords:** sarcoma, surgery, classification, risk, margins

## Abstract

**Simple Summary:**

In soft tissue sarcomas the width of surgical margins after resection determines the extent of surgery and the function after resection. But how far is really necessary? 305 patients with deep-seated, G2/3 soft tissue sarcomas of the extremity, the trunk wall, or the pelvis were reviewed. The 5-year local recurrence-free survival (LRFS) was 82%. Overall survival (OS) at 5 years was 66%. Positive (contaminated) margins worsened LRFS and OS. A margin of >10 mm did not improve LRFS and OS as compared to one of >5 mm. A resection margin of <1 mm showed a trend but not significantly better LRFS or OS compared to a contaminated margin. In conclusion the margin should at least be free of tumor, in sound tissue. A margin of >5 mm sound tissue seems to be sufficient. Resecting more tissue does not benefit the patient.

**Abstract:**

Background: The significance of surgical margins after resection of soft tissue sarcomas in respect to local-recurrence-free survival and overall survival is evaluated. Methods: A total of 305 patients with deep-seated, G2/3 soft tissue sarcomas (STS) of the extremity, the trunk wall, or the pelvis were reviewed. The margin was defined according to the Fédération Nationale des Centres de Lutte Contre le Cancer (FNCLCC) classification system (R0-2), the Union Internationale Contre le Cancer (UICC) classification (R + 1 mm) for which a margin <1 mm is included into the R1 group, and in groups of <1 mm, 1–5 mm, >5 mm, or >10 mm. Results: Of these patients, 31 (10.2%) had a contaminated margin, 64 (21%) a margin of <1 mm, 123 (40.3%) a margin of 1–5 mm, 47 (15.4%) a margin of >5 mm, and 40 (13.1%) a margin of >10 mm. The 5-year local recurrence-free survival (LRFS) was 81.6%. Overall survival (OS) at 5 years was 65.9%. Positive margins worsened LRFS and OS. A margin of >10 mm did not improve LRFS and OS as compared to one of >5 mm. Conclusions: A resection margin of <1 mm showed a trend but not significantly better LRFS or OS compared to a contaminated margin. This finding supports use of the UICC classification. A margin of more than 10 mm did not improve LRFS or OS.

## 1. Introduction

An adequate, and in general wide, resection is the standard method for treating soft tissue sarcomas [1]. Limb-sparing procedures have been recommended for this purpose since at least 1985 by an National Institute of Health (NIH) consensus statement [2]. If diagnosed, local recurrence is an independent prognostic factor for overall survival (OS) [3,4,5,6,7,8]. However, the prognostic significance of surgical margins remains controversial [9]. In a meta-analysis of 33 studies by Kandel et al. in 2013, 21 studies demonstrated a negative impact of positive margins on local recurrence (LR), with only one study not showing that [10]. Overall survival was influenced by positive margins in only one study and no difference was seen in three others as for example in a large study published by the Scandinavian Sarcoma Group [11].

The 2020 National Comprehensive Cancer Network (NCCN) guidelines acknowledge that close margins may be necessary to preserve such critical structures as vessels and nerves. Close soft tissue margins as defined by these guidelines are <1 cm [12]. The European Society of Medical Oncology (ESMO) Clinical Practice Guidelines in 2018 recommend: “a wide excision with negative margins (no tumor at the margin, R0)” [13].

Hence, the aim of this study was to bring more profound data into this discussion of the influence of margin width not only on LR but also on OS. Due to a comparatively-large number of high-grade sarcoma patients, most of most of whom had been treated surgically with a curative approach in a recent 5-year period by only two surgeons within a homogenous neo- and adjuvant interdisciplinary setting over the whole period and with prospectively-registered resection margins, we expected clinically-relevant results.

## 2. Patients and Methods

Between 2012 and 2017, 305 patients with deep-seated G2 and G3 tumors of the extremities, the trunk wall, or the pelvis had surgery.

Prior to surgery, magnetic resonance imaging (MRI) and, in select cases, computed tomography (CT) were used to define size and localization of the tumor. A CT scan of the chest was performed to determine the presence or absence of pulmonary metastatic disease. This was repeated at follow-up. Prospectively, the margin status of the resection was evaluated.

The margin was defined according to the Fédération Nationale des Centres de Lutte Contre le Cancer (FNCLCC) grading system as being R0 if a layer of healthy tissue around the lesion was present (wide resection) or R1 if the margins were contaminated but the tumor capsule remained closed (marginal resection). In select patients, a planned partial resection was performed in order to avoid severely-mutilating surgery. This situation was classified as an R2 resection. In all cases of R0 resections, we grouped the patients into four categories—margins of <1 mm, 1–5 mm, >5 mm, or >10 mm. Margin thickness was assessed by measuring the smallest diameter of the surrounding cuff of healthy tissue in the formaldehyde-fixed specimens.

### 2.1. Chemotherapy

A total of 162 (53%) patients received chemotherapy (CTX). It was administered either preoperatively or pre- and postoperatively. Neoadjuvant, multi-agent therapy consisted mostly of AI (Adriamycin and Ifosfamid), EIA (Etoposid, Ifosamid, and Adriamycin), and other regimens in some instances [14,15]. Local hyperthermia was added in many cases.

### 2.2. Radiotherapy

In addition, 212 (70%) patients received neoadjuvant or adjuvant radiotherapy (RTX). Preoperative RTX was applied at a dose of 50 Gy. Adjuvant external beam radiation was applied at a dose of 60–66 Gy.

### 2.3. Histopathology

All histopathologic samples of biopsies or resected tumors were reviewed by the same pathologist (TK). The distribution of histotypes is summarized in Table 1.

### 2.4. Endpoints and Statistics

In this prospective evaluation, the patients were analyzed with regards to local and distant tumor spread and with the main end-points being local recurrence-free survival (LRFS) and OS. All patients were followed for evidence of LR or distant metastasis. LRFS and OS were defined either as the time from surgery to the first occurrence of LR or to death from any cause. For statistical analysis, OS and LRFS were calculated according to the Kaplan–Meier method. Significance analysis was performed using the log-rank or the Cox proportional-hazards regression model. A *p* value of less than 0.05 was considered statistically significant. The data analysis software used was MedCalc^®^ (MedCalc Software, Ostend, Belgium).

### 2.5. Ethics Approval and Consent to Participate

This study was approved by the ethics committee of the Medical Faculty, University of Munich (17-891). Written consent was obtained from all the patients included in this study.

## 3. Results

### 3.1. Patient Characteristics

The median age of the 164 male and 141 female patients was 63.4 years (mean 60.5, range 5–99). The median tumor size was 5.7 cm (mean 7.1, range 0–32). Of these patients, 236 (77%) had a primary sarcoma and 69 (23%) had recurrent disease. In addition, 32 (11%) patients had metastatic disease at the time of diagnosis.

A wide (R0) resection was performed in 274 (89.8%) cases, a marginal (R1) resection in 27 cases (8.9%), and an R2 resection in 4 (1.3%) patients. Evaluation of the margins showed that 31 (10.2%) patients had a contaminated margin, 64 (21%) a margin of <1 mm, 123 (40.3%) a margin of 1–5 mm, 47 (15.4%) a margin of > 5mm, and 40 (13.1%) a margin of >10 mm. Histology resulted in G2 tumors in 121 (39.7%) and G3 tumors in 184 (60.3%) cases.

Twelve (3.9%) patients were lost to follow-up less than 12 months after surgery and 75 (24.6%) patients deceased during follow-up.

### 3.2. Metastatic Disease, Local Recurrence-Free Survival, and Overall Survival

The 5-year LRFS was 81.6%; 40 (13.1%) patients developed local recurrence. At final follow-up, 87 (28.5%) patients had metastatic disease. Overall survival of the entire group at 5 years was 65.9%.

### 3.3. Influence of Grading, Margins, and Local Recurrence on LRFS and OS

Positive margins had a highly-significant negative impact on LRFS (Figure 1, *p* = 0.0194); the better the margin that could be obtained, the lower the recurrence rate (Figure 2, *p* = 0.0045). However, a margin of more than 10 mm did not increase the LRFS compared to one of more than 5 mm.

For OS, grading was a highly-significant predictive factor (*p* = 0.0006) with a 5-year OS of 73.6% vs. 60.7% for G2 and G3 sarcomas, respectively. Local recurrence itself also significantly worsened OS (Figure 3, *p* = 0.0018) from 69.4% to 45.9% after 5 years. The margin status in general significantly influenced OS (Figure 4, *p* = 0.0005). Excluding the R2-resected sarcoma patients, however, significance was lost (*p* = 0.2281), but in detailed analysis of the margin the same as in LRFS was obvious. A significantly-better OS was seen as margins increased until >5 mm, but >10 mm did not improve OS further (Figure 5, *p* = 0.0288). After excluding the R2-resected sarcomas the *p*-value dropped to *p* = 0.0621.

In multivariate analysis, R status, grading, size, age, and RTX had a significant influence on OS (Table 2).

A comparison of the major factors in 305 patients with soft tissue sarcoma in respect to the detailed resection margins is shown in Table 3.

## 4. Discussion

The 5-year LRFS and OS in this study correlated well with other large studies [17,18,19]. A contaminated surgical margin enhanced the risk of local recurrence. Over the years, different residual tumor classification systems have been established. The R classification based on the microscopic (R0/1) and also macroscopic (R2) evaluation is the best-established of these. In 2002, the UICC proposed a R + 1 mm classification, for which a margin of less than 1 mm (<1 mm) sound tissue at any part of the specimen is included into the R1 group [20]. In our experience, LRFS between true R1 resections and resections of <1 mm was different in trend. However, this difference failed to reach statistical significance. Gundle et al. compared both systems in 2018 [21]. They concluded that the traditional R classification best determined the risk of LR in a competing risk framework. They also investigated the Toronto Margin Context Classification (TMCC) [22,23] in which planned positive margins at critical structures are distinguished from unplanned positive margins. This latter system may help to further classify the risk of LR in positive margins.

McKee at al., could show, that in three groups with margins of 0, 1-9, and ≥10 mm, LRFS was only longer in the last group but with no impact on OS [24]. In one of the recent retrospective studies (*n* = 305, G2/3 only) [25] the authors could not find differences between the <1 mm, <2 mm, <5 mm or <10 mm groups. But it was a limitation of their study that not all negative margins could retrospectively be sorted into the right category.

The selection of patients is a major contributing factor for a detailed analysis. If STS of all gradings (G1–3) and locations (deep or superficial) and over a longer period of time with changing treatment concepts regarding RTX and CTX are included, the results might not be concise. So in some studies, the favorable impact of negative margins on LFRS and OS is the best that could be evaluated [26,27,28,29]. Even contaminated margins may not influence OS at all [30]. If the patient selection is more specific (deep, high-grade) the significance of positive margins on LR is clear [31]. Studies showing that margins have no impact on LR are exceedingly rare. Most authors have concluded that RTX might contribute to this effect [32].

Radiotherapy has a major impact on LR after R0 or marginal resections and can decrease the risk of LR significantly [31,33]. Radiotherapy was applied in this study in 70% of the patients; 74% of the patients with contaminated margins, 75% of those with margins of <1 mm, 76% of those with margins of 1–5 mm, 72% with margins of 5–10 mm, and 33% of patients with margins of ≥10 mm received RTX. RTX was an independent positive factor for LRFS and OS. In the relevant subgroups with up to <10 mm resection margins RTX was approximately equally distributed.

As Gundle et al., also described [21] we saw a difference between a contaminated margin and margins of <1 mm, but this was not significant (Figure 2). Regarding OS our observation was the same (Figure 4). However, in contrast and in extension to Gundle et al., the R0 group in our study showed significant differences between patients with <1 mm, 1–5 mm, and > 5 mm margins. If all patients with ≥1 mm free margins were grouped into one cohort and compared to those with contaminated margins and those with margins of <1 mm, a 5-year LRFS of 68% (contaminated), 70% (<1 mm), and 87% (≥ 1 mm) was seen (*p* = 0.0011). That means that patients with a very close margin of <1 mm in fact had a worse prognosis than those with larger free margins in respect to LRFS. For OS this effect was visible as a trend but not statistically significant. In their comment to Gundle et al., Levy et al. discussed the heterogeneity of their study group [34].

In a study comparing the R and R+1 classification, an LR was seen in 9.5% of R0-resected patients and in 36.7% of R1-resected patients after 5 years [35]. Using the UICC R + 1 mm, an LR of 12% was seen in R0 + 1 resected patients vs. 36.9% in R1 patients after 5 years [35]. However, due to non-registered resection margins measurements (mm), more than 70% of the patients in the study group had to be excluded. The authors concluded that patients with a resection margin of <1 mm have the same prognosis as those with positive resection margins. Lintz et al. examined the same in 2011 [36]. They saw a 5-year LRFS of 90% after R0- and of 64% after R1 resection, compared to 92% after R0 + 1- and 64% after R1 + 1 resection. They also came to the conclusion that a margin of <1 mm is comparable to a contaminated margin. Disease-free survival significantly deteriorated for positive margins in this study. Due to similar observations in their own patients, many authors consider a margin of <1 mm as a positive margin [18].

In our study, OS was influenced by a positive margin. This is even clearer if the numbers of patients studied are higher (Stojadinovic et al., *n* = 2084 [5]). In smaller studies even with a homogenous patient cohort, the statistical power may not be sufficient to show this (Vraa et al, *n* = 152 [9]).

In 1371 soft tissue sarcoma patients, O’Donnell et al. described 169 cases with positive margins [22]. They did not only look for the quality of the margins but also for the clinical setting in which they occurred. Unexpected positive margins showed the worst outcome for LRFS and DSS. Critical structure positive margins (as anticipated on vessels or nerves to avoid amputation) had no significant influence on LRFS. Thus, they concluded that avoiding resection of critical structures by accepting locally-close margins seemed to be safe.

Willeumier et al., in 2015 showed a significant impact of margin width (contaminated, ≤2 mm, and >2 mm) on LRFS but not on OS in 127 patients, whereas not many other studies could prove an effect of positive margins on OS [29].

Potter et al., in 2013 showed this highly significantly in a retrospective analysis on 363 patients with soft-tissue sarcomas and positive margins [37]. Our own multivariate analysis also proved the negative influence of positive margins on OS. LR itself was not a factor (Table 2). Interestingly, if analyzed with respect to DSS in the study of Potter et al., LR also lost its significance. Biau et al. in 2012 could show that an increased risk of LR and also reduced OS was associated with positive margins [38]. In their analysis of competing risks, grading, tumor size, and deep/superficial location lost significance.

In our study, a margin of ≥10 mm did not improve LRFS or OS more than a margin of 5–10 mm. Even more, a margin of ≥10 mm showed an inferior effect on OS that was between the effects of a margin < 1 mm and a contaminated margin. Liu et al. in 2010 retrospectively examined 181 patients including all grades and locations [39]. They divided the groups into contaminated, 1–4 mm, 5–9 mm, 10–19 mm, 20–29 mm, and ≥30 mm margins. Contaminated margins did worse on LRFS, 1–4 mm and 5–9 mm margins had about the same risk of about 40% LR after 5 years, and patients with ≥10 mm margins had nearly no risk of LR. This was also seen (<10 mm vs. ≥10 mm) in respect to DSS. They concluded that 10 mm or more should be an adequate margin. One has to mention that in this study, the rate of LR was comparatively high, even for negative margins of 5 mm or more. Their conclusion might in fact have been influenced by the selection of the patients or by the surgical technique.

In the study of Dickinson et al., in 2006 on 279 patients, margins of 1–4 mm, 5–9 mm, and 10–19 mm showed about the same rate of LR [40]. Margins of <1 mm or contaminated margins proved to be worse. Our own data with a very homogenous patient cohort showed about the same but was also able to differentiate better between the 1–4 mm, 5–9 mm, and the >10 mm groups.

Novais et al., in 2010 investigated 248 patients with high grade (G2–G4) sarcomas [41]. They divided them into four groups (contaminated, ≤2 mm, >2 mm to ≤2 cm, and >2 cm margins), but for statistical reasons clustered the contaminated and ≤2 mm group. They showed not only a higher rate of LR after 5 years in the first group (11.6% vs. 2.4% vs. 0%) but also an inferior 5-year OS of 89.1% vs. 93.5% vs. 100%. A margin of between 2 mm and 2 cm was as good as one of more than 2 cm. LR and age and stage of the disease did significantly influence OS. The disadvantage of that study is the comparatively-large margin interval in the second group which is of high clinical interest.

### Radiotherapy

Radiotherapy has been reported to convert a marginal margin to a negative margin [42,43,44]. In a study dating back to 1994/95 in G2/3 extremity sarcomas, 50 Gy neoadjuvant RTX was administered [45]. The 5-year LRFS was 91% in the R0 and 62% in the R1/2 groups and the 5-year OS was not significantly different. Kim et al. in 2010 published a series of 56 patients with STS of all grades; 6 had a positive and 14 a marginal (<1 mm) margin [46]. All patients received on median 50 Gy neoadjuvant RTX, 12 patients a second boost postoperatively. LR was seen in three of the six R1 lesions and in two of the 14 marginal resections. In this setting, a margin of ≥1 mm seemed to be adequate for the authors. Regarding the postoperative boost in patients with a positive surgical margin who already had a neoadjuvant RTX (50 Gy), a boost of 16 Gy did not provide any advantage in preventing LR [47].

## 5. Limitations of the Study

Soft tissue sarcoma patients are always included in a multimodal treatment strategy. Hence the evaluation of a single factor as margins, divided into subgroups as in this study, may be affected by other factors such as radiation therapy. Multi-variant analysis did not account properly for an indication bias in this cases. RTX, for example, is part of our concept in tumors with close resection margins, with CTX in large G2/3 sarcomas excluding patients with high age. Hence margin, tumor size, and age are also inadvertently linked to RTX and CTX.

As in some other studies, patients with metastatic disease at initial diagnosis but with tumors judged to be resectable, were included.

We did not analyze the quality of the margins, as for example when cutting through muscle tissue as opposed to an anatomic border like a fascia or periosteum. It is clear that anatomical borders have a preventive effect on tumor growth and LR after marginal resection [48]. We tried a classification including such anatomic borders at the beginning of the margin registration but soon came into trouble. An anatomic margin in nearly all of the cases is only in part seen at the specimen. Furthermore, if local recurrence occurs it is impossible to determine which part of the resection specimen caused it.

## 6. Conclusions

This study shows that a resection margin of <1 mm provides only a non-significant advantage for LRFS and OS over a contaminated margin. This speaks in favor of using the UICC classification there R0 is defined as a margin of ≥1 mm. We were able to demonstrate a difference between less or more than a 5 mm margin width, but a margin of more than 5–10 mm did not improve LRFS or OS. Radiotherapy had an independent effect on both LRFS and OS.

## Figures and Tables

**Figure 1 cancers-12-02560-f001:**
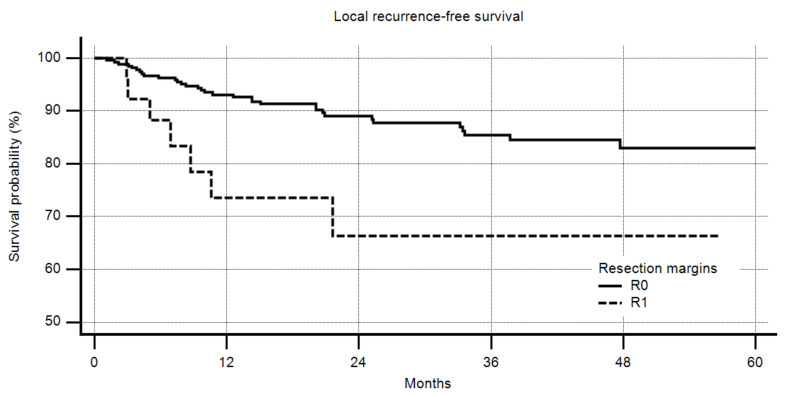
Local recurrence-free survival by margin (R0 vs. R1), R2 excluded, *n* = 301, 5-year R0 83%, 5-year R1 66%, *p* = 0.0194.

**Figure 2 cancers-12-02560-f002:**
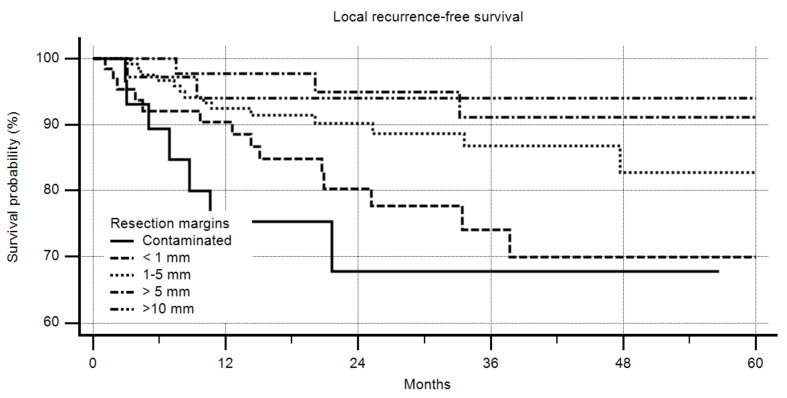
Local recurrence-free survival by margin in detail, *n* = 305, *p* = 0.0045. In multivariate analysis, only age, adjuvant radiotherapy (RTX), and chemotherapy (CTX) had a significant impact on local recurrence-free survival (LRFS) (Table 2).

**Figure 3 cancers-12-02560-f003:**
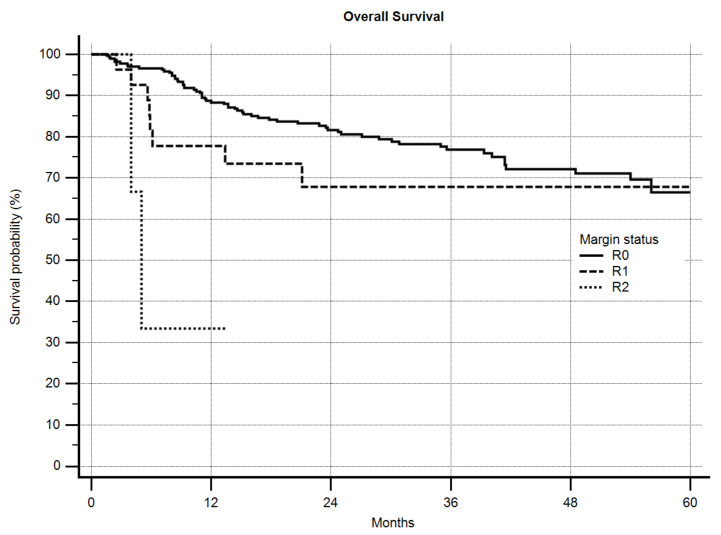
Overall survival by local recurrence, *n* = 305, *p* = 0.0018.

**Figure 4 cancers-12-02560-f004:**
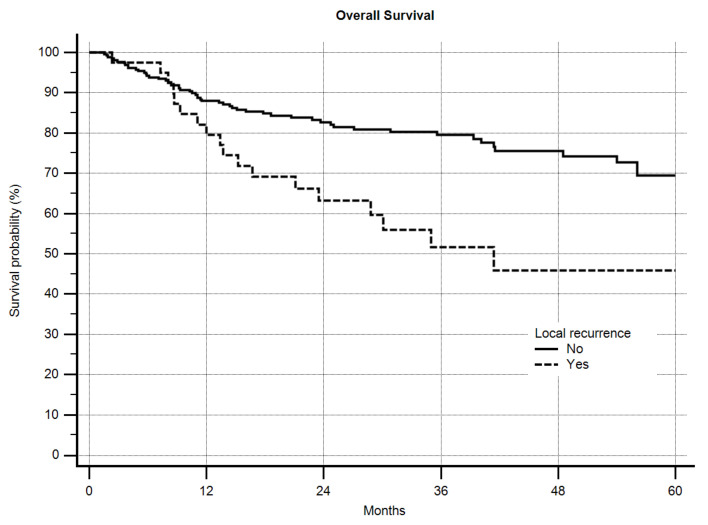
Overall survival by margin (R0, R1, R2), *n* = 305, *p* = 0.0005.

**Figure 5 cancers-12-02560-f005:**
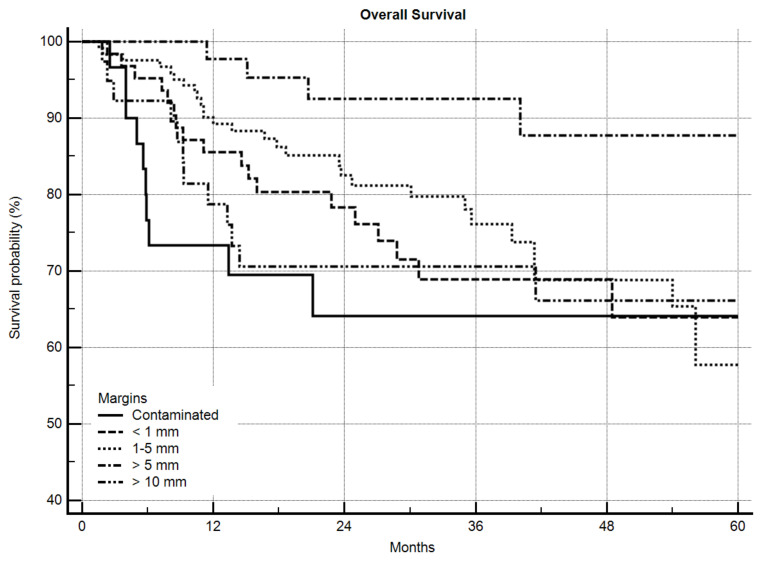
Overall survival by margin in detail, *n* = 305, *p* = 0.0288.

**Table 1 cancers-12-02560-t001:** Comparison of local recurrence, tumor size, and grading by histology in 305 patients with soft tissue sarcomas.

	LR	Tumor Size (cm, Median)	G2	G3	All
All patients	40 (13.1%)	5.7	121 (39.7%)	184 (60.3%)	305 (100%)
Undifferentiated pleomorphic sarcoma (UPS)	12 (9.9%)	7.0	18 (15%)	103 (85%)	121 (40%)
Myxofibrosarcoma	6 (16.2%)	5.5	19 (51%)	18 (49%)	37 (12%)
Liposarcoma	1 (2.9%)	7.5	23 (66%)	12 (34%)	35 (11%)
Leiomyosarcoma	5 (21.7%)	6.0	9 (39%)	14 (61%)	23 (8%)
Synovial sarcoma	0 (0%)	3.0	13	7	20 (7%)
Malignant peripheral nerve sheath tumor (MPNST)	6 (33.3%)	4.9	10 (56%)	8 (44%)	18 (6%)
Fibrosarcoma	6 (40%)	3.5	11 (73%)	4 (27%)	15 (5%)
Epitheloid sarcoma	0 (0%)	3.7	3 (50%) *	3 (50%) *	6 (2%)
Clear cell sarcoma	0 (0%)	4.6	2 (33%) *	4 (67%) *	6 (2%)
Rhabdomyosarcoma	1 (20%)	5.0	3 (60%) *	2 (40%) *	5 (2%)
Other rare sarcomas	3 (15.8%)	5.1	10 (53%) *	9 (47%) *	19 (6%)

* Grading of alveolar rhabdomyosarcoma, angiosarcoma, clear cell sarcoma, and epithelioid sarcoma is in general not recommended; for this study we used an adaption to the Coindre grading system [16].

**Table 2 cancers-12-02560-t002:** Factors influencing local recurrence-free survival (LRFS) and overall survival (OS). Multivariate Cox proportional-hazards regression in 305 patients.

Covariate	B	SE	Wald	P	Exp(b)	95% CI of Exp(b)
**Local Recurrence-Free Survival (LRFS)**						
Margin (R0–R2)	0.6732	0.4375	2.3675	0.1239	1.9605	0.8316 to 4.6216
Grading	0.1978	0.3470	0.3248	0.5688	1.2187	0.6173 to 2.4059
Size	0.03842	0.02562	2.2489	0.1337	1.0392	0.9883 to 1.0927
Age	0.02759	0.01223	5.0898	**0.0241**	1.0280	1.0036 to 1.0529
RTX	−0.7286	0.3399	4.5958	**0.0321**	0.4826	0.2479 to 0.9395
CTX	1.2659	0.3974	10.1493	**0.0014**	3.5464	1.6276 to 7.7274
**Overall Survival (OS)**						
Margin (R0–R2)	0.7858	0.3362	5.4638	**0.0194**	2.1942	1.1353 to 4.2408
Grading	0.8264	0.2879	8.2401	**0.0041**	2.2850	1.2997 to 4.0173
Size	0.03930	0.01858	4.4718	**0.0345**	1.0401	1.0029 to 1.0787
Age	0.01776	0.008437	4.4333	**0.0352**	1.0179	1.0012 to 1.0349
LR	0.4611	0.2874	2.5734	0.1087	1.5858	0.9028 to 2.7855
RTX	−1.2379	0.2403	26.5433	**<0.0001**	0.2900	0.1811 to 0.4644
CTX	0.2311	0.2695	0.7356	0.3911	1.2600	0.7430 to 2.1367

B, unstandardized regression weight; SE, variation unstandardized regression weight; WALD, Wald test; P, probability P (bold ≤0.05); Exp(b), hazards ratio; 95% confidence interval for hazards ratio.

**Table 3 cancers-12-02560-t003:** Comparison of clinical factors in 305 patients with soft tissue sarcoma in respect to detailed resection margins.

	Contaminated	<1mm	1–5 mm	>5 mm	>10 mm	All
**All patients**	31 (10.2%)	64 (21%)	123 (40.3%)	47 (15.4%)	40 (13.1%)	305 (100%)
**Age**						
<60 years	13 (9.8%)	21 (15.8%)	54 (40.6%)	24 (18.0%)	21 (15.8%)	133 (43.4%)
≥60 years	18 (10.5%)	43 (25.0%)	69 (40.1%)	23 (13.4%)	19 (11.0%)	172 (56.4%)
**Sex**						
Female	11 (7.8%)	30 (21.3%)	63 (44.7%)	20 (14.2%)	17 (12.1%)	141 (46.2%)
Male	20 (12.2%)	34 (20.7%)	60 (36.6%)	27 (16.5%)	23 (14.0%)	164 (53.8%)
**Tumor size**						
<5 cm	3 (2.4%)	15 (11.8%)	55 (43.3%)	33 (26.0%)	21 (16.5%)	128 (42.0%)
≥5 cm	28 (15.8%)	49 (27.7%)	68 (38.4%)	14 (7.9%)	18 (10.2)	177 (58.0%)
**Location**						
Upper extremity	10 (13.9%)	17 (23.6%)	26 (36.1%)	9 (12.5%)	10 (13.9%)	72 (23.6%)
Trunk/pelvis	2 (4.8%)	6 (14.3%)	15 (35.7%)	10 (23.8%)	9 (21.4%)	42 (13.8%)
Lower extremity	19 (9.9%)	41 (21.5%)	82 (42.9%)	28 (14.7%)	21 (11.0%)	191 (62.6%)
**UPS histology**						
Yes	9 (7.4%)	24 (19.8%)	58 (47.9%)	18 (14.9%)	12 (9.9%)	121 (39.7%)
No	22 (12.0%)	40 (21.7%)	65 (35.3%)	29 (15.8%)	28 (15.2%)	184 (60.3%)
**Grading**						
G2	15 (12.4%)	24 (19.8%)	44 (36.4%)	23 (19.0%)	15 (12.4%)	121 (39.7%)
G3	16 (8.7%)	40 (21.7%)	79 (42.9%)	24 (13.0%)	25 (13.6%)	184 (60.3%)
**Radiotherapy**						
Yes	23 (10.8%)	48 (22.6%)	94 (44.3%)	34 (16.0%)	13 (6.1%)	212 (69.5%)
No	8 (8.6%)	16 (17.2%)	29 (31.2%)	13 (14.0%)	27 (29.0%)	93 (30.5%)
**Chemotherapy**						
Yes	21 (13.0%)	36 (22.2%)	66 (40.7%)	23 (14.2%)	16 (9.9%)	162 (53.1%)
No	10 (7.0%)	28 (19.6%)	57 (39.9%)	24 (16.8%)	24 (16.8%)	143 (46.9%)

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
