# Peer review of "The Effect of Resection Margin on Local Recurrence and Survival in High Grade Soft Tissue Sarcoma of the Extremities: How Far Is Far Enough?"

_cancers, 2020, doi:10.3390/cancers12092560_

Round 1
Reviewer 1 Report
In this manuscript the authors evaluate the importance of 2 classifications used to define margin status after resection of soft tissue sarcomas from extremity, pelvis and trunk wall. The authors conclude that < 1mm margins should be considered as contaminated and are associated with high risk of local relapse and worse overall survival. On the other hand margins > 100 mm did not improve overall survival.
Comments/questions as follows:
In figure 5 the OS of > 10mm margin seems inferior to >5 mm margin. How do the authors explain that?
Chemotherapy appears to have impact on LRFS but not on OS. Can be a discussion point?
The breakdown of location of soft tissue sarcomas should be given. Achieving negative margins in pelvic location tumors can be quite challenging without doing reconstructions or amputations. Outcomes in this specific subgroup should be looked at closely and discussed.
Reviewer 2 Report
Dear Editor.
Annika Bilgeri et al, in their paper analyzed the effect of resection margin on local recurrence and survival. However some statistical data and tables need to be revised.
Table 2 please substitute comma with point and insert abbreviation in legend
Table1 please collect the clinical pathological characteristics of patients in a single table and indicate the recurrences, tumor size and grade by histology (Tables 1-3)
It is unclear how many local recurrence patients were analyzed.
From table 5. margin >10mm showed the same effect of contaminated margins Please could the author discuss why CTX affects LRFS and not OS and radiation influenced only OS
Author Response
Dear Editors,
Thank you very much for reviewing this manuscript. We think that the comments of both reviewers will increase the clarity of this study.
As proposed by the reviewers we made the following corrections:
Reviewer #2:
- Table 2: please substitute comma with point and insert abbreviation legend.
Done as proposed.
- Table 1: please collect the clinical pathological characteristics of patients in a single table and indicate the recurrences, tumor size and grade by histology (Tables 1-3).
Table 1 was revised as proposed and reads now:
“Table 1. Comparison of local recurrence, tumor size and grading by histology in 305 patients with soft tissue sarcomas.”
- It is unclear how many local recurrence patients were analyzed.
On page 6 it reads:
“Metastatic disease, local recurrence-free survival and overall survival
5-year LRFS was 81.6%. 40 (13.1%) patients developed local recurrence. At final follow-up, 87 (28.5%) patients had metastatic disease. Overall survival of the entire group at 5 years was 65.9%.”
So 40 patients with LR had been analyzed. Due to the changes asked for by this reviewer for the new table 1, this very important point should have now more awareness to the reader.
- From table 5: margin >10 mm showed the same effect of contaminated margins.
This is a very important point as also mentioned by reviewer #1. This is true. Nearly the same effect as contaminated margins.(n.s.). As written above we checked that twice in our data. To be honest, we do not have an explanation for that. This group of patients, as shown in the table, is not different in respect to grading, size and age to the group of 1-5 mm.
We think this non expected result in OS (in great difference to LRFS) is part of the discussion regarding margins and OS at all. To emphasize that, we included a sentence in the discussion on page 15. Here it reads before and after:
“In our study, a margin of ≥10 mm did not improve LRFS or OS more than a margin of 5-10 mm.”
We added: “Even more, a margin of ≥10 mm showed an inferior result on OS in between those of a margin < 1 mm and a contaminated margin.”
- Please could the author discuss why CTX affects LRFS and not OS and radiation influenced only OS.
Regarding CTX reviewer #1 made the same comment. We wrote to his remark:
“In this cohort of patients chemotherapy was distributed very unevenly. So many older patients did not receive CTX. For that any discussion on that point lacks sufficient data. We included two references of our latest publication to CTX in this group of STS patients (14 and 15) which might be read by the interested reader. But thank both reviewers for this observation.”
Regarding RTX (Table 4):
You are right, this is not understandable. There are a number of studies regarding RTX which prove the effectiveness. We think this is at least to a great part caused by the univariate analysis. RTX was given in selected patients, those with close margins and large tumours. So the indication itself is a bias. To compensate for that, one has to have a look at the univariate analysis (Table 3). Here RTX is significantly associated with LRFS as also OS. After a second internal discussion we think, table 4 does not add any more information to table 3 and hence we deleted that table.
Thank you once again for reviewing this manuscript!
Round 2
Reviewer 2 Report
Dear Editor, the authors addressed all my points
in my opinion, the manuscript is acceptable for publication